# Aicardi-Goutières syndrome-associated gene SAMHD1 preserves genome integrity by preventing R-loop formation at transcription–replication conflict regions

Kiwon Park[1,2☯], Jeongmin Ryoo[3☯], Heena Jeong[1,2], Minsu Kim[1,2], Sungwon Lee[1,2], Sung-Yeon Hwang[1,2], Jiyoung Ahn[4], Doyeon Kim[2], Hyungseok C. Moon[5], Daehyun Baek[2], Kwangsoo Kim[6], Hye Yoon Park[5], Kwangseog Ahn[1,2]*

1 Center for RNA Research, Institute for Basic Science, Seoul, Republic of Korea, 2 School of Biological Sciences, Seoul National University, Seoul, Republic of Korea, 3 Department of Hematology, Oncology and Stem Cell transplantation, Comprehensive Cancer center Freiburg, University of Freiburg, Freiburg, Germany, 4 Biomedical Research Institute, Seoul National University Hospital, Seoul, Republic of Korea, 5 Department of Physics and Astronomy, Seoul National University, Seoul, Republic of Korea, 6 Transdisciplinary Department of Medicine & Advanced Technology, Seoul National University Hospital, Seoul, Republic of Korea

☯ These authors contributed equally to this work.
* ksahn@snu.ac.kr

**Data Availability Statement:** The bioinformatics data and figures reported in this study are available under GitHub (https://github.com/spiritmage72/

## Abstract

The comorbid association of autoimmune diseases with cancers has been a major obstacle to successful anti-cancer treatment. Cancer survival rate decreases significantly in patients with preexisting autoimmunity. However, to date, the molecular and cellular profiles of such comorbidities are poorly understood. We used Aicardi-Goutières syndrome (AGS) as a model autoimmune disease and explored the underlying mechanisms of genome instability in AGS-associated-gene-deficient patient cells. We found that R-loops are highly enriched at transcription-replication conflict regions of the genome in fibroblast of patients bearing SAMHD1 mutation, which is the AGS-associated-gene mutation most frequently reported with tumor and malignancies. In SAMHD1-depleted cells, R-loops accumulated with the concomitant activation of DNA damage responses. Removal of R-loops in SAMHD1 deficiency reduced cellular responses to genome instability. Furthermore, downregulation of SAMHD1 expression is associated with various types of cancer and poor survival rate. Our findings suggest that SAMHD1 functions as a tumor suppressor by resolving R-loops, and thus, SAMHD1 and R-loop may be novel diagnostic markers and targets for patient stratification in anti-cancer therapy.

## Author summary

Mutations in SAMHD1 cause Aicardi-Goutières syndrome (AGS), a monogenic lupus-like autoimmune disease. Among AGS-associated genes, SAMHD1 is most frequently mutates in various types of tumors and malignancies, suggesting that it is biologically

samhd1_rloop). The modified custom python scripts used to determine DRIP-seq signal across HO and CD regions were made available under GitHub (https://github.com/spiritmage72/samhd1_rloop). All relevant underlying numerical data for all graphs and summary statistics are either in Spreadsheets of Supporting information or included in relevant data.

**Funding:** This work was supported by the Institute for Basic Science of the Ministry of Science Grant (IBS-R008-D1) and the National Research Foundation of Korea grant funded by the Korea government (NRF-2020R1A5A1018081) (to K. A.), (NRF-2020R1A2C3011298) (to K.A. and K.P.), and the BK21 plus fellowship (A0423-20200100) (to K. P.). The funders had no role in study design, data collection and analysis, decision to publish, or preparation of the manuscript.

**Competing interests:** The authors have declared that no competing interests exist.

relevant to cancer development. Here, we show that SAMHD1 resolves R-loops induced by transcription-replication conflicts, thereby contributing to the maintenance of genome stability. Our findings provide insight into the molecular and mechanical understanding of the autoimmunity and cancer comorbidity, and suggest that SAMHD1 and R-loops are potential and reliable biomarkers in anti-cancer therapeutics.

## Introduction

Recent advances in immunotherapy have led to the development of more efficient cancer therapeutics. However, the autoimmune comorbidities with cancer are the major obstacles to successful anti-cancer treatment, particularly in patients with preexisting autoimmune disease (AD). Despite of the greater risk of developing cancer in patients with preexisting AD than in the general population [1–3], cancer patients with preexisting AD have been excluded from immunotherapy and clinical trials because of concerns about the exacerbation of their autoimmune responses [4]. Consequently, to date, cancer survival rate has been decreasing significantly in patients with preexisting AD [5]. Therefore, the conditions in which cancer arises in AD call for the development of novel therapeutic options based on a better understanding of the molecular and cellular profiling of such comorbidities.

Most of autoimmune diseases have complex genetic factors and do not show a Mendelian pattern of inheritance. Therefore, it is challenging to elucidate the mechanism underlying the observed comorbidity between AD and cancer. Aicardi-Goutières syndrome (AGS) is a rare monogenic AD whose clinical importance is magnified as its clinical and biochemical characteristics highly overlap with much common AD, systemic lupus erythematosus (SLE) [6]. AGS mimics the congenital infection and it exhibits type I interferonopathy, which is considered central to the pathogenesis of the SLE [7]. AGS is caused by mutations in genes including *SAMHD1*, *TREX1*, *RNASEH2A*, and *RNASEH2B*, most of which play roles in nucleic acid metabolism [8]. Deficiencies in AGS-associated genes often share the common features of abnormal activation of the immune system due to aberrant nucleic acid metabolism. However, among AGS-associated genes, sterile alpha motif and HD domain-containing protein 1 (SAMHD1) is most frequently reported in the context of multiple cancer types and malignancies [9–14]. Therefore, SAMHD1-associated AGS provides an excellent experimental model for studying the comorbidity of AD and cancer.

SAMHD1 exhibits dual enzymatic activities: deoxynucleoside triphosphohydrolase (dNTPase) and phosphorolytic $3'$-$5'$ -exoribonuclease activities [15–17]. SAMHD1 is frequently mutated not only in AGS but also in a variety of human cancers, such as chronic lymphocytic leukemia, colorectal cancer, and epidermotropic cutaneous T-cell lymphoma [10,13,18]. Additionally, downregulation of SAMHD1 expression has been reported in cutaneous T-cell lymphoma and lung adenocarcinoma [11]. Intriguingly, the recently reported role of SAMHD1 in DNA repair and replication fork processing upon genotoxic reagent treatment is independent of its known enzymatic activity, that is, dNTPase activity [19,20]. Accumulating evidence indicates that SAMHD1 plays a role in preserving genome integrity and preventing cancer. However, the mechanisms by which SAMHD1 preserves genomic integrity against intrinsic DNA damage and by which an endogenous source causes cancer development in SAMHD1-deficiency but not in other AGS-associated gene deficiencies remain unknown.

R-loops, also known as DNA:RNA hybrids, are transcription byproducts that play crucial roles in several biological processes, such as immunoglobulin class switching and epigenetics [21]. Dysregulation of R-loops and the resulting harmful R-loops cause replication fork

stalling, which could serve as a source of genome instability [22]. R-loops can be formed during transcription-replication conflicts (TRCs) within gene body regions and that they induce distinct DNA damage responses depending on the orientation of collisions between the DNA replisome and RNA polymerase [23]. Head-on (HO) collisions occur when the DNA replisome and RNA polymerase II (RNAP II) process toward each other and collide in the "HO" direction and they exacerbate the formation of genome-destabilizing R-loops [23]. On the other hand, co-directional (CD) TRCs occur when the replication and transcription machineries proceed along the same DNA template at different rates as the DNA replisome and RNAP II move in the same direction [24], and resolves R-loops. Interestingly, R-loops accumulate in fibroblasts of patients with AGS [25]. AGS-specific R-loops are quite different from the conventional R-loops, which are observed at the transcription start site, transcription termination site, and GC skew. Fibroblasts with mutations in different AGS-associated genes exhibit unique patterns of R-loop accumulation along genomic regions. However, the characteristics and biological relevance of these R-loops remains elucidated.

Genome instability is a hallmark of cancer and R-loop enrichment around TRCs has been suggested as a potent threat to genome integrity. In this study, we analyzed genome-wide DNA:RNA hybrid accumulation at TRC regions in AGS patient fibroblasts via bioinformatics analysis based on multiple databases. R-loop formation at TRC regions was found only in the fibroblasts of AGS patients bearing SAMHD1 mutations but not in those of other AGS-associated gene mutations. Despite the absence of exogenous DNA damaging stimuli, we found that SAMHD1-deficient cells exhibit R-loop-mediated genome instability. Furthermore, analyses of The Cancer Genome Atlas (TCGA) database indicated that downregulation of SAMHD1 expression occurs in various cancer types and is associated with poor survival in colorectal cancer patients. Our study identifies an unexpected function of SAMHD1 in resolving R-loops that accumulate following TRCs, thereby contributing to the maintenance of genome stability and preventing cancer.

## Results

### R-loops adjacent to replication origins accumulate in fibroblasts of AGS patients bearing SAMHD1 mutations

To investigate the relationship between SAMHD1 and R-loops that give rise to genome instability, we analyzed deposited DNA:RNA hybrid immunoprecipitation followed by the sequencing (DRIP-seq) data obtained from AGS patient fibroblasts with deficiencies in SAMHD1 and other AGS-associated genes [25]. We focused on the accumulation of R-loops in gene body regions, particularly with replication origins, where TRCs potentially occur. To distinguish gene body regions with or without replication origins, we employed an existing dataset that maps replication origins in actively transcribed genes [23,26–28]. This dataset was established by integrating published Okazaki fragment sequencing (OK-seq) data that defines replication origins in the human genome and global run-on sequencing (GRO-seq) data to identify transcribed genes [23]. Normalized DRIP-seq reads were mapped to each gene body region: origins in gene bodies (n = 727) or centers of gene bodies (n = 484), where replication origins were present or absent, respectively. This enabled us to predict R-loop accumulation at genomic regions with replication origins in 24 kb windows (±12 kb windows centered on replication origins of gene bodies) and distinguish TRCs as HO or CD based on their collision orientations. We therefore compared R-loop accumulation levels across each gene body region in different types of fibroblasts from patients with AGS. Remarkably, SAMHD1-deficient AGS (AGS5) patient fibroblasts exhibited a much-enriched DRIP-seq read-count in comparison with fibroblasts from healthy controls, in gene body regions with replication origins (Fig 1A,

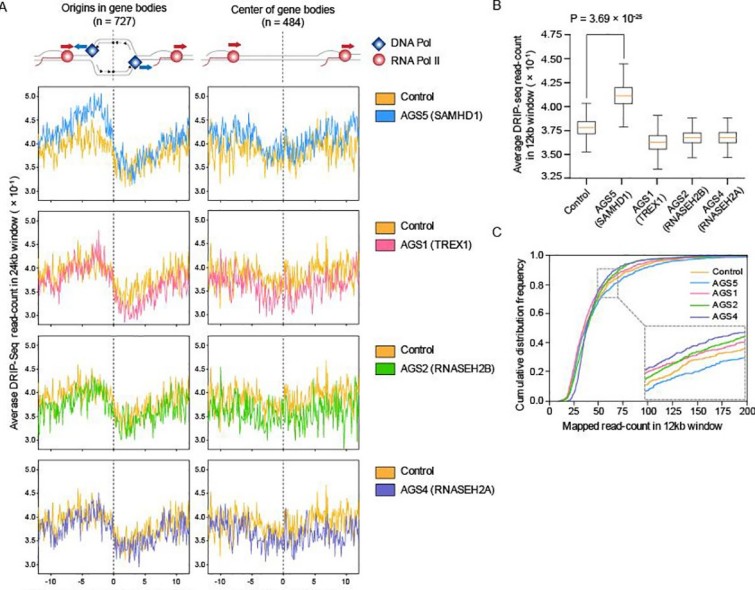

**Fig 1. SAMHD1 mutations induce R-loop formation by transcription–replication conflict in AGS patient fibroblasts. A,** Average mapped DRIP-seq read accumulation around replication origins across the gene bodies (n = 727) (left) or center of the same gene bodies (n = 484) (right) in fibroblasts from control (yellow) and AGS patients with the indicated gene mutations (blue, SAMHD1; pink, TREX1; green, RNASEH2B; purple, RNASEH2A). For individual graph, the line indicates the mean DRIP-seq read-count of each genomic region of 24 kb windows centered on origins or center of gene bodies. **B,** Box graph indicating quantified average DRIP-seq read-count across 12 kb window centered on the replication origins of the gene bodies (n = 727). For each sample, orange line in the center of boxes indicates the median, and the box indicates 1st and 3rd quartiles. Statistical significance was assessed using the Fisher's exact test followed by Bonferroni correction. **C,** Cumulative distribution frequency of DRIP-seq mapped read-count around each replication origins of the gene bodies in 12 kb windows centered on replication origins of the gene bodies (n = 727). Individual line represents the cumulative distribution frequency of healthy control and AGS patients with the indicated gene mutations (yellow, healthy control; blue, AGS5; pink, AGS1; green, AGS2; purple, AGS4).

left), but not in the control region of the same gene bodies without replication origins (Fig 1A, right). However, patient fibroblasts with deficiencies in other AGS-associated genes, such as TREX1 (AGS1), RNASEH2A (AGS4), and RNASEH2B (AGS2), exhibited similar levels of R-loop enrichment along both replication origins and centers of gene bodies with that of the healthy control (Fig 1A, right and left). To eliminate any bias in our bioinformatics data processing, we mapped DRIP-seq reads on random genomic regions covering approximately 40% of the whole genome, and none of the AGS patient fibroblast samples exhibited DRIP-seq read-count enrichment in these negative control regions (S1A Fig). Since such AGS-associated genes are involved in nucleic acid metabolism [8], their mutation could have altered the global transcriptome and thereby genomic TRC profile. Therefore, we have analyzed RNA sequencing (RNA-seq) data obtained from the fibroblasts of AGS patients and compared them with those of healthy control [25]. Among the selected genes we used for DRIP-seq analysis, there were only a few of differentially expressed genes (DEGs) in AGS patient fibroblasts (S1B Fig). The average DRIP-seq read-count around the DNA replication regions of gene bodies in 12 kb windows (±6 kb windows centered on replication origins of gene bodies) was quantified. R-loops in TRC regions were significantly enriched only in SAMHD1-deficient fibroblasts when compared to that of the fibroblasts from healthy control (adjusted P = 3.69x10$^{-25}$, Fisher's exact test); however, the enrichment was not observed in other AGS-associated gene-deficient fibroblasts (Fig 1B). In addition, cumulative distribution frequency curves indicated that

fibroblasts bearing the SAMHD1 mutation had the highest proportion of enriched DRIP-seq read mapping at each 12 kb window along the region around the origins (Fig 1C). Interestingly, the DRIP-seq read-count at HO collision sub-regions was highly and significantly enriched in SAMHD1-deficient fibroblasts (adjusted P = $6.5 \times 10^{-33}$, Fisher's exact test), while other AGS associated gene-deficient fibroblasts showed insignificant differences compared to that of healthy control (S1C Fig). On the other hand, SAMHD1-deficient fibroblasts showed insignificant DRIP-seq read-count enrichment at CD collision sub-regions (adjusted P = 0.0707, Fisher's exact test), while other AGS associated gene-deficient fibroblasts showed even lower R-loop accumulation in comparison to that of healthy control (S1D Fig). HO collisions, which occur when the DNA replisome and RNAP II collide in a head-on orientation, exacerbate the formation of genome-destabilizing R-loops, which activate DNA damage responses [23]. Instead, CD-oriented TRCs resolve R-loops [23]. These results therefore suggest that SAMHD1 deficiency promotes TRC-dependent R-loop accumulation and that these R-loops could be a potent threat to normal replication fork progression and genome integrity.

## SAMHD1 deficiency induces replication stress and delays S phase progression

We depleted SAMHD1 protein expression in human U2OS cells with short hairpin RNA (shRNA) expression vectors targeting SAMHD1 untranslated regions (UTRs) (S2A Fig). We examined the resulting cellular phenotypes in terms of genome instability in SAMHD1-deficiency without any genotoxic agent treatment, which exogenously induced replication stress or double-strand DNA breaks. As TRCs and abnormal accumulation of R-loops can lead to replication fork stalling and slowing of replication, we examined the proliferation of SAMHD1-knockdown cells. SAMHD1-depleted U2OS cells grew much slower than the rate at which control cells transfected with a Luciferase-targeting shRNA vector grew (S2B Fig). However, this defect in cell growth was not a result of apoptosis or cell cycle arrest (S2C, S2D and S2E Fig). Given that there was no arrest of the overall cell cycle, we examined cell cycle progression in SAMHD1-depleted cells in detail using a BrdU pulse-chase assay to determine whether there was a cell cycle delay. Unsynchronized cells were pulse-labeled with BrdU and allowed to continue cycling. Compared with control cells, only approximately half of the cells entered the next G1 phase when SAMHD1 was depleted, at 9 h after the removal of BrdU (Fig 2A and 2B). Despite the temporal or spatial separation between the replication and transcription of some genes, the two cellular processes necessarily occur on the same DNA template at the same time under certain circumstances [29]. Accordingly, TRCs inevitably occur in the S phase and cause genome instability when or where there are deficiencies in the temporal or spatial safeguards [24,30,31]. Replication stress restrains cell cycle progression, particularly when it is perceived by cell cycle checkpoint proteins [27]. This suggests that cells experiencing a high level of replication stress from TRCs and R-loops would exhibit slow S phase progression. S phase progression was analyzed in order to confirm that SAMHD1-depleted cells undergo DNA replication stress and perturbation during cell cycle progression. Cells were synchronized to the G1/ early S phase using a double thymidine block and then released to continue cycling (S3A Fig). The population in each phase of the cell cycle was determined by propidium iodide (PI) staining followed by flow cytometry analysis. SAMHD1-knockdown cells exhibited much slower S phase progression than control cells, as indicated by a higher percentage of the population entering the S phase at later time points (8 h and 10 h post-release) (Fig 2C). Considered together, these data indicate the depletion of SAMHD1 protein results in intrinsic replication stress, which in turn leads to defects in S phase progression.

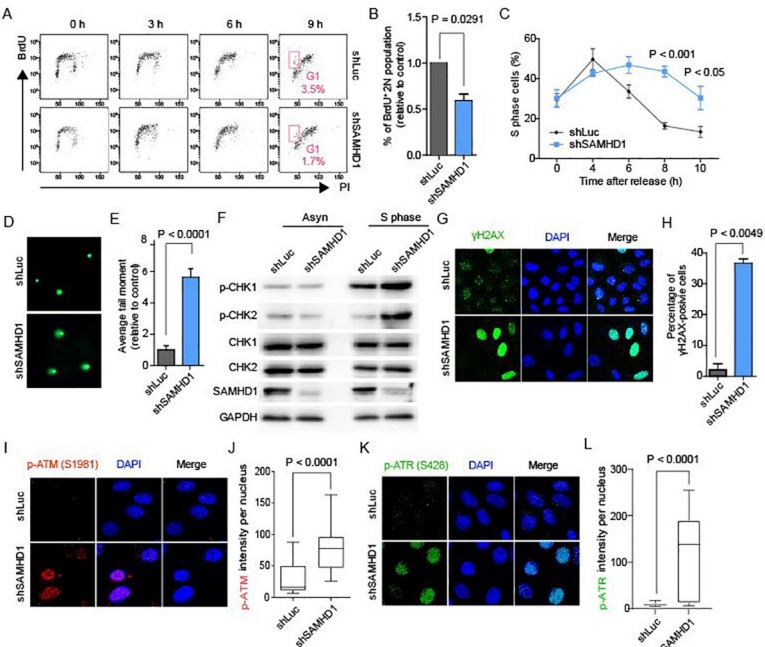

**Fig 2. Depletion of SAMHD1 induces cell cycle delay and spontaneous DNA damage response. A,** Representative flow plot of SAMHD1-depleted or control U2OS cells in BrdU pulse-chase experiment showing G1 phase cell population after BrdU labelling (pink box). **B,** Quantification analysis of BrdU pulse-chase experiment in **A**. Data represent mean ± SEM percentages of BrdU positive G1 phase (2N) cell population. Statistical significance was assessed using two-tailed Student's $t$-test (n = 3). **C,** Flow cytometry prolife of SAMHD1-depleted and control U2OS cells at different time points after release from double thymidine block. Cells were subjected to PI staining followed by FACS analysis. Data represent mean ± SEM, n = 3, P values were calculated using two-way ANOVA. **D,** Representative images of comet assay performed under alkaline electrophoresis condition in SAMHD1-depleted and control U2OS cells. **E,** Quantification analysis of comet tail moment lengths for each condition showed in **D**. Data represent mean ± SEM, the statistical significance was assessed using two-tailed Student's $t$-test (n = 50). **F,** Immunoblots of CHK1 phosphorylated on S345 (p-CHK1), CHK2 phosphorylated on T68 (p-CHK2), and CHK1 and CHK2 in asynchronized (Asyn) or S phase-synchronized (S phase) SAMHD1-depleted and control U2OS cells (n = 2). **G,** Representative image of immunofluorescence assay of H2AX phosphorylated on S139 (γH2AX) in S phase synchronized SAMHD1-depleted and control U2OS cells. Cells were co-stained with DAPI (blue) and anti-γH2AX antibodies (green) (n = 2). **H,** Quantification of γH2AX positive signal per nucleus for the immunofluorescence assay described in **G**. Data represent mean ± SEM, the statistical significance was assessed using the two-tailed Student's t-test (n >5). **I,** Representative image of immunofluorescence assay of ATM phosphorylated on S1981 (p-ATM) in S phase-synchronized SAMHD1-depleted and control U2OS cells. Cells were co-stained with DAPI (blue) and anti-p-ATM antibodies (red) (n = 2). **J,** Quantification of p-ATM positive signal per nucleus for the immunofluorescence assay described in **I**. The box plot indicates values from minimum to maximum, the statistical significance was assessed using the Mann-Whitney $U$ test (n >100). **K,** Representative image of immunofluorescence assay of ATR phosphorylated on S428 (p-ATR) S phase synchronized in SAMHD1-depleted and control U2OS cells. Cells were co-stained with DAPI (blue) and anti-p-ATR antibodies (green) (n = 2). **L,** Quantification of p-ATR positive signal per nucleus for the immunofluorescence assay described in **I**. The box plot indicates values from minimum to maximum, the statistical significance was assessed using the Mann-Whitney $U$ test (n >100).

## SAMHD1 is required for preventing DNA damage in non-stressed cells

As the DNA damage response is accompanied by replication stress [27] and TRC-dependent R-loops [23], we examined whether the depletion of SAMHD1 induces DNA damage and genome instability. We tested the overall genome integrity in SAMHD1-depleted U2OS cells using an alkaline single cell gel electrophoresis (SCGE) assay, which is also known as the comet assay. The comet assay under alkaline condition enables a simple evaluation of cellular DNA damage, including single- and double-stranded DNA breaks, the majority of apurinic/apyrimidinic sites, and alkali-labile DNA adducts. SAMHD1-depleted U2OS cells harbored

more DNA damage than control cells, as indicated by the formation of longer tail moments (Fig 2D and 2E). Moreover, the length of comet tail moment in SAMHD1-depleted cells decreased following the reconstitution of SAMHD1 protein using a SAMHD1-HA-expressing construct (S3B and S3C Fig). A particularly interesting aspect of this observation was that DNA damage occurred spontaneously without any exogenous stimuli following the loss of SAMHD1. This suggests that SAMHD1 depletion leads to an accumulation of endogenous sources of genome instability.

Further, we examined the phosphorylation levels of CHK1 and CHK2 in both asynchronized and S-phase-synchronized SAMHD1-knockdown and control cells. CHK1 and CHK2 are the replication checkpoint proteins and substrates of the ATR and ATM checkpoint kinases, respectively. Because TRC-dependent R-loops can appear during the S phase [29–31], and S phase progression was slowed down in SAMHD1-depleted cells (Fig 2C), we tested the activation levels of CHK1 and CHK2 in the S phase-synchronized cell population. The SAMHD1-depleted and control U2OS cells were evaluated 4 h after release from the double-thymidine cell cycle block, which is a widely used DNA replication inhibitor with relatively low cytotoxicity [32] (S3D Fig). The phosphorylation levels of both CHK1 and CHK2 were increased in S phase-synchronized SAMHD1-depleted cells but not in the heterogeneous cell population (Fig 2F). This result is in line with a very recent report that demonstrated S-phase specific DNA damage response in cells experiencing TRC-mediated R-loop accumulation [33]. This could explain why the spontaneous activation of DNA damage response signaling proteins in SAMHD1-deficient cells could not be observed in previous studies [10,19,20,34]. Moreover, the activation statuses of CHK1 and CHK2 were restored by ectopic SAMHD1 protein expression (S3E and S3F Fig). Consistent with this finding, S phase-synchronized SAMHD1-knockdown cells exhibited increased activation of H2AX, a major component of the DNA damage response, the phosphorylation of which serves as a sensitive marker of DNA damage (Fig 2G and 2H). HO-oriented TRCs promote ATR activation, which responds to ssDNA, while CD-oriented TRCs promote ATM activation, which is primarily activated by DSBs [23]. R-loop accumulation in SAMHD1-deficient fibroblasts at HO and CD collision sub-regions were different (S1C and S1D Fig). Therefore, we examined the phosphorylation levels of ATM and ATR in S phase-synchronized SAMHD1-depleted and control U2OS cells. SAMHD1-knockdown cells exhibited increased activation of both ATM (p-ATM) (Fig 2I and 2J) and ATR (p-ATR) (Fig 2K and 2L). Since cellular R-loop processing may convert an R-loop to a DSB [35], this result is in agreement with a previously proposed molecular mechanism of R-loop-mediated DNA damage response. Additionally, we have observed longer tail moments in asynchronized SAMHD1-depleted cells than those in control cells under neutral condition, which enables an evaluation of cellular double-stranded DNA breaks (S3G and S3H Fig). Overall, these data suggest that the depletion of SAMHD1 protein gives rise to obstacles to the maintenance of genome stability, such as TRC-dependent R-loops.

## SAMHD1 prevents cellular R-loop accumulation

R-loops are considered as endogenous threats to genomic integrity [21,22], and their accumulation has been observed in tissues of cancer patients [36]. Following the bioinformatics approach that revealed TRCs-dependent R-loop accumulation in SAMHD1-deficient AGS fibroblasts (Fig 1), we analyzed cellular TRCs and R-loop accumulation following SAMHD1 depletion in S phase-synchronized U2OS cells. First, we examined whether SAMHD1-depletion in U2OS cells increases TRCs by a proximity-ligation assay (PLA) between RNA polymerase II (RNAPII) and PCNA, which are essential proteins in the transcription and replication machineries, respectively. The RNAPII-PCNA PLA signal was increased upon SAMHD1

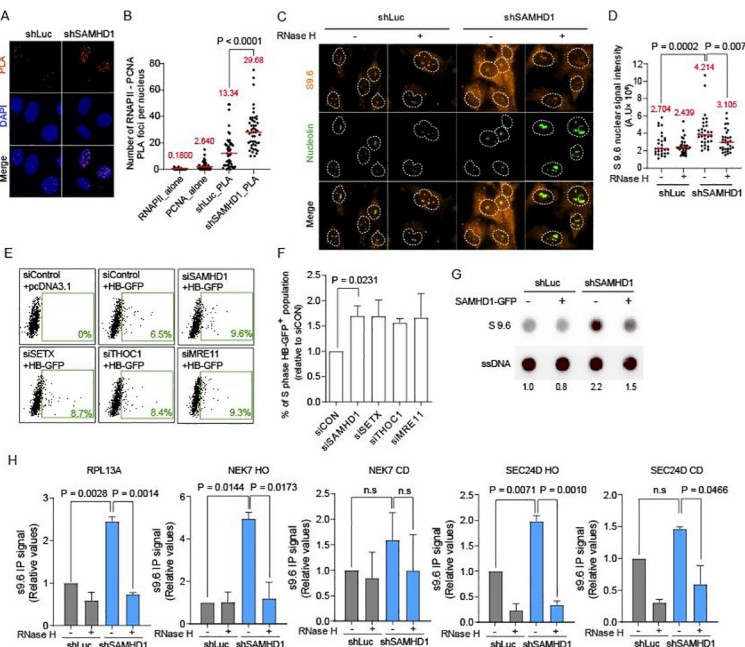

**Fig 3. Depletion of SAMHD1 causes cellular R-loop accumulation. A,** Representative images of proximity ligation assay (PLA) between PCNA and RNA polymerase II (RNAPII) in S phase-synchronized SAMHD1-depleted and control U2OS cells. Cells were subjected to PLA (red) and co-stained with DAPI (blue). **B,** Quantification analysis of number of PLA foci per nucleus in **A**. RNAP II_alone and PCNA_alone are single-antibody controls from S phase-synchronized SAMHD1-depleted U2OS cells. The mean value for each data is shown as a red line. Statistical significance was assessed using the two-tailed Mann-Whitney $U$ test (n = 50). **C,** Representative images of immunofluorescence assay of S9.6 nuclear signals in SAMHD1-depleted and control U2OS cells. Cells were co-stained with anti-S9.6 (orange) and anti-nucleolin antibodies (green). The nucleus (defined by DAPI image) is outlined. Where indicated, cells were incubated with or without RNase H in vitro before staining with anti-S9.6 antibodies. **D,** Quantification of S9.6 signal intensity per nucleus after nucleolar signal subtraction for the immunofluorescence assay described in **C**. The mean value for each data is shown as a red line. Statistical significance was assessed using one-way ANOVA (n >28). **E,** Representative fluorescence-activated cell sorting (FACS) profile of HB-GFP retention assay, examining basal R-loop levels in U2OS cells treated with control (siControl) or indicated siRNAs.pcDNA3.1 transfected siControl-treated U2OS cells are negative control for HB-GFP retention assay. **F,** Quantification analysis of HB-GFP retention assay co-stained with PI. Data represent mean ± SEM percentages of GFP positive S phase cell population. Statistical significance was assessed using two-tailed Student's $t$-test (n = 3). **G,** Dot blot analysis of R-loop in genomic DNA extracts from SAMHD1-depleted and control U2OS cells, in the absence or presence of ectopic SAMHD1-GFP expression (n = 3). The genomic DNA were probed with anti-S9.6 and anti-ssDNA antibodies to verify equivalent DNA loading. Fold-induction was normalized to the control U2OS cells in absence of ectopic SAMHD1-GFP expression by quantifying the dot intensity of the blots and calculating the ratios of S9.6 signal to total amount of genomic DNA. **H,** DRIP-qPCR using the anti S9.6 antibody at RPL13A, NEK7 HO, NEK7 CD, SEC24D HO and SEC24D CD are shown in SAMHD1-depleted and control U2OS cells. Pre-immunoprecipitated samples were untreated (-) or treated (+) with RNase H, as indicated. Values are relative to those of control U2OS cells. Data represent mean ± SEM the statistical significance was assessed using the one-way ANOVA (n = 2).

depletion (Fig 3A and 3B). SAMHD1-knockdown or control cells were probed with S9.6 antibody, which specifically binds to DNA:RNA hybrids and detects R-loops. We observed R-loop accumulation in SAMHD1-depleted cells by DNA:RNA hybrid immunofluorescence assay using the S9.6 antibody, and the S9.6 signal (orange) intensity per nucleus was quantified after removal of the nucleolar signal (green) (Fig 3C). The nuclear fluorescence signal associated with R-loops was enhanced in cells depleted of SAMHD1 (Fig 3D). We also measured R-loop levels using an HB-GFP retention assay [37]. HB-GFP is a fusion protein consisting of the DNA:RNA hybrid-binding (HB) domain of RNaseH1 tagged with EGFP, which enables detection of R-loops in cells using flow cytometry analysis of GFP-associated fluorescence. We performed HB-GFP retention assay upon small interfering RNA (siRNA)-mediated depletion of

SAMHD1 and other well-known R-loop regulators [21,38], in asynchronized U2OS cells (S4A Fig). Cells were transiently transfected with an HB-GFP-expressing vector for the HB-GFP retention assay, which quantifies only R-loop-binding HB-GFP signals following cell semi-permeabilization and washing away of proteins that do not bind R-loops (S4B Fig). To observe the HB-GFP-positive S phase cell population, we co-stained cells with PI. Flow cytometry analysis showed that SAMHD1, SETX, THOC1 and MRE11 knockdown led to an increase in the HB-GFP-positive population (Fig 3E), whereas HB-GFP transfection efficiency and expression levels were not affected by target gene knockdown (S4C Fig). Moreover, the levels of R-loop accumulation in S phase cells were enriched in SAMHD1-, SETX-, THOC1- and MRE11-knockdown cells in comparison to those in control cells (Fig 3F), while the proportion of S phase cells was not affected by target gene knockdown (S4D Fig). The R-loop accumulation in SAMHD1-depleted cells was also observed by DNA:RNA hybrid dot blots using the anti-S9.6 antibodies, dot intensity declined when SAMHD1 protein was restored (Figs 3G and S4E). The enhanced R-loop signals on dot blots of SAMHD1-knockdown cells decreased significantly upon *in vitro* treatment with RNase H, which specifically degrades RNAs of DNA:RNA hybrids (S4F Fig). We further confirmed R-loop accumulation in SAMHD1-depleted cells and validated our bioinformatics analysis using DNA-RNA immunoprecipitation and qPCR (DRIP-qPCR) using anti-S9.6 antibodies. In this case, the S9.6 signal was determined for genes actively transcribed such as NEK7 and SEC24D, and we used RPL13A as a positive control, which has been previously validated for the detection of R-loops [37]. Additionally, we further specified R-loop accumulating regions in NEK7 and SEC24D gene bodies into the HO collision sub-region (NEK7 HO, SEC24D HO) and CD collision sub-region (NEK7 CD, SEC24D CD) based on the dataset that maps replication origins in actively transcribed genes [23]. First, we confirmed that the depletion of SAMHD1 does not affect the genomic TRC profile, as transcription levels of the subjected genes are not affected by SAMHD1 protein depletion (S4G Fig). We assumed that the replication origin usages are not altered by SAMHD1 protein depletion, as it is previously observed in AGS fibroblasts and HEK293T cell line [19,39]. We detected the accumulation of R-loops at the RPL13A, NEK7 HO and SEC24D HO regions in SAMHD1-depleted cells compared to those in the control U2OS cells, whereas at the NEK7 CD and SEC24D CD did not show significant R-loop accumulations (Fig 3H). Importantly, *in vitro* RNase H treatment induced a dramatic signal decrease at RPL13A, NEK7 HO, and SEC24D HO regions, confirming that enriched signals were R-loop specific (Fig 3H). Moreover, we have investigated more number of genes such as GBE1, TRIO, MAP4K4, CALD1 and PALM2-AKAP2. The transcription levels of those genes were not altered by SAMHD1 protein depletion (S4H Fig). As expected, the DRIP signals on HO regions of all those genes were enriched by more than 1.5-fold in SAMHD1-depleted cells compared to that of control cells (S4I Fig, top). However, DRIP signals on CD regions of the genes were comparable in SAMHD1-deplted cells and control cells (MAP4K4 and PALM2-AKAP2), or slightly increased in SAMHD1-depleted cells but with larger adjusted P values than their DRIP signals on HO regions (GBE1, TRIO and CALD1) (S4I Fig, bottom). Together, these results indicate that SAMHD1 prevents R-loop formation, which strongly agrees with our bioinformatics analysis.

## SAMHD1 regulates R-loop-mediated genome instability

A correlation between SAMHD1-dependent R-loop accumulation and the cellular DNA damage response has not been previously described. To further test our hypothesis that TRC-dependent R-loops are the endogenous source of genome instability in SAMHD1-deficiency, we rescued SAMHD1-depleted cells by RNaseH1, which specifically resolves R-loops. The slower S phase in SAMHD1-depleted cells was rescued when cells were transiently transfected

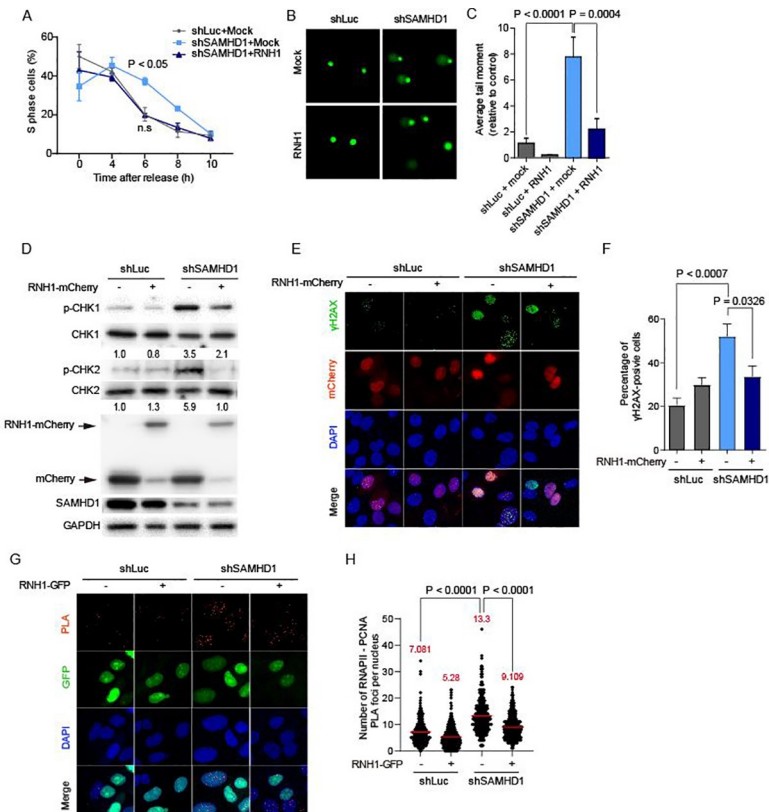

**Fig 4. Depletion of SAMHD1 induces R-loop-mediated genome instability and cellular DNA damage response. A,**
Flow cytometry prolife of control U2OS and SAMHD1-depleted cells, in the absence or presence of ectopic
RNaseH1-GFP expression at different time points after release from double thymidine block. Cells were subjected to PI
staining followed by FACS analysis. Data represent mean ± SEM, n = 3, P values were calculated using two-way
ANOVA. **B,** Representative images of comet assays performed under alkaline electrophoresis conditions in
SAMHD1-depleted and control U2OS cells, in the absence or presence of ectopic RNaseH1-mCherry expression. **C,**
Quantitative analysis of comet tail moment lengths for each condition described in **B**. Statistical significance was
assessed using two-tailed Student's *t*-test (n = 50). **D,** Immunoblots of CHK1 and CHK2 in S phase-synchronized
SAMHD1-depleted and control U2OS cells, in the absence or presence of ectopic RNaseH1-mCherry
(RNH1-mCherry) expression. Fold-induction was normalized to the empty vector-transfected control U2OS cells by
quantifying the band intensity of the blots and calculating the ratios of p-CHK1 and p-CHK2 to total expression levels
of CHK1 and CHK2 expression, respectively. **E,** Representative image of immunofluorescence assay of H2AX
phosphorylated on S139 (γH2AX) in S phase synchronized SAMHD1-depleted and control U2OS cells, in the absence
or presence of ectopic RNaseH1-mCherry (RNH1-mCherry) expression. Cells were co-stained with DAPI (blue) and
anti-γH2AX antibodies (green) (n = 2). **F,** Quantification of γH2AX positive signal per nucleus for the
immunofluorescence assay described in **E**. Data represent mean ± SEM, the statistical significance was assessed using
the two-tailed Student's t-test (n > 5). **G,** Representative images of proximity ligation assay (PLA) between PCNA and
RNA polymerase II (RNAPII) in S phase synchronized SAMHD1-depleted and control U2OS cells, in the absence or
presence of ectopic RNaseH1-GFP (RNH1-GFP) expression (n = 2). Cells were subjected to PLA (red) and co-stained
with DAPI (blue). **H,** Quantification analysis of number of PLA foci per nucleus in **G**. The mean value for each data is
shown as a red line. Statistical significance was assessed using the two-tailed Mann-Whitney *U* test (n = 322).

with a construct encoding wild-type *E. coli* RNaseH1 (RNH1) (Fig 4A). The level of DNA
damage in SAMHD1-depleted cells was evidenced by the longer tail moment in the SCGE
assay. The degree of DNA damage was also reduced by ectopic expression of RNH1 (Fig 4B
and 4C). Moreover, the DNA damage response in S phase-synchronized SAMHD1-depleted
cells recovered upon WT RNase H1 expression, as indicated by reduced phosphorylation levels
of both CHK1 and CHK2 (Fig 4D). Consistent with this, increased activation of H2AX in the
absence of SAMHD1 was also rescued by RNH1 (Fig 4E and 4F). Further, we examined

whether TRCs in SAMHD1-depleted cells could be recovered by RNH1. The PCNA-RNAPII PLA signal that was enriched in SAMHD1-knockdown cells was reduced by the ectopic expression of RNH1 (Fig 4G and 4H). Overall, these results reveal that the TRC-dependent R-loops serve as the primary source of replication stress leading to genome instability in SAMHD1-deficiency.

## Endogenous SAMHD1 expression is associated with cancer

Genome instability is a hallmark of cancer, and dysregulated R-loops are a great threat to genome integrity [22]. Because we identified R-loops as an endogenous source of genome instability in SAMHD1 deficiency, we focused on the association between SAMHD1 expression and cancer development. We analyzed The Cancer Genome Atlas (TCGA) database to evaluate the SAMHD1 expression levels in various cancer types. RNA sequencing by Expectation Maximization (RSEM) analysis of SAMHD1 expression levels in different cancer types revealed that SAMHD1 expression was downregulated in tumor tissues (Fig 5A, red dots) compared to that in normal control tissues (Fig 5A, green dots). This agrees with previous reports of downregulated SAMHD1 expression in chronic lymphocytic leukemia, cutaneous T-cell lymphoma, and lung adenocarcinoma [10,11,13,18]. Consistent with our TCGA database analysis, we have identified that some of colorectal cancer-relevant SAMHD1 mutants [40] have lower expression levels than that of the wild-type SAMHD1 (S5A and S5B Fig). SAMHD1-deficient AGS fibroblasts (AGS5) exhibited a distinct R-loop formation pattern compared to fibroblasts with other AGS-associated gene deficiencies (Fig 1). Even though AGS-associated gene deficiencies often share common physiologic features and disorders caused by type I interferonopathies, malignancy has most frequently been reported in the

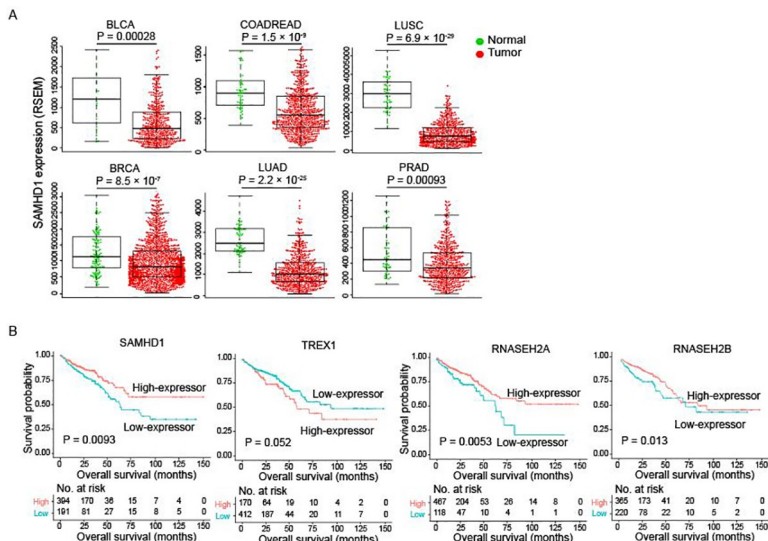

**Fig 5. SAMHD1 is downregulated in many cancer types and associated with cancer patient survival rate. A,** Box and whisker plots showing RSEM (RNA-Seq by Expectation Maximization) analyses of SAMHD1 expression levels in several cancer types (red) versus those in the corresponding normal tissues (green) in TCGA datasets (LUSC: lung squamous cell carcinoma, LUAD: lung adenocarcinoma, BLCA: bladder urothelial carcinoma, READ: rectal adenocarcinoma, COAD: colon adenocarcinoma, BRCA: breast invasive carcinoma, PRAD: prostate adenocarcinoma). The median, quartiles and range of values are shown. Statistical significance was assessed using log-rank test. **B,** Kaplan-Meier plots for the overall survival of SAMHD1, TREX1, RNASEH2A, and RNASEH2B high-expressor (red) and low-expressor (blue) groups of COADREAD patients. COADREAD data were obtained from TCGA database, and the cut-off was estimated by the optimal cut-point. Statistical significance was assessed using the log-rank test. The P values for Kaplan-Meier analyses of each genes are indicated.

context of SAMHD1 [10,11,14]. Thus, we examined the correlation between the R-loop accumulation patterns in the deficiency of each AGS-associated gene with cancer. The overall survival rate was compared between colorectal adenocarcinoma (COADREAD) patients with high versus low AGS-associated gene expression levels. Interestingly, patients exhibiting low expressions of SAMHD1 and RNASEH2A (Fig 5B, blue) had a significantly lower survival rate in comparison with that in patients exhibiting high SAMHD1 and RNASEH2A expressions (Fig 5B, red). However, there was no significant difference in patient survival rate between patients exhibiting high versus low expressions of TREX1 and RNASEH2B, which did not show distinct R-loop signal accumulation in DRIP-seq analysis (Fig 1A). RNASEH2B has been reported as a colorectal tumor suppressor in its knockout mouse model, but in P53 dependent manner [41]. RNASEH2A silencing causes reduced proliferation and increased apoptosis of human glioblastoma cells *in vitro* [42]. This observation could explain why low RNASEH2A-expressiong COADREAD patients had a low overall survival rate despite insignificant R-loop signal enrichment in RNASEH2A-mutated AGS patient fibroblasts. To test the relevance of the TRC-dependent R-loop regulatory function of SAMHD1 in cancer, we have examined whether colorectal cancer-relevant SAMHD1 mutants can regulate cellular TRCs. We identified two SAMHD1 mutants (F59C and T232M), which could be involved in cancer development due to their inabilities to regulate normal TRC-dependent R-loop regulation (S5C and S5D Fig).

## Discussion

In this study, we set to understand the molecular mechanism and pathologies underlying the comorbidities of AD and cancer using a well-known model of systemic autoimmune disorder, AGS. An elegantly designed bioinformatics study has demonstrated that R-loop enrichment around TRCs is a potent threat to genome integrity [23]. The approach of that study inspired us to investigate whether TRC-dependent R-loop formation serves as an endogenous source of genome instability in AGS, a model disease for systemic autoimmunity. We integrated various bioinformatics data, and analyzed and compared DRIP-seq data regarding AGS patient fibroblasts. We found that R-loops are highly enriched at regions near replication origins in gene bodies only in SAMHD1-mutated AGS patient fibroblasts and that SAMHD1-deficient fibroblasts exhibited a significantly higher genome-destabilizing R-loops derived from HO TRCs. Our results are consistent with a previous report that SAMHD1 localizes to replication foci and increased fork stalling was observed by DNA fibre assay in SAMHD1- knockdown cells [19]. In line with these results, we observed that S-phase progression was delayed, and DNA damage responses were spontaneously activated in SAMHD1-deficiency cells. Importantly, we confirmed that TRCs and R-loop accumulation are enhanced in SAMHD1-deficient cells, and SAMHD1-deficient cells exhibit R-loop-mediated genome instability. Notably, by analyzing the TCGA dataset, we demonstrated that SAMHD1 transcription is downregulated in many cancer types. Moreover, the overall survival rate of COADREAD patients was found to be significantly lower in patients with low SAMHD1 expression.

SAMHD1 is an AGS-associated gene, that is reported to be frequently mutated in multiple types of cancer [14]. In addition, our TCGA analysis results and previous reports demonstrate downregulation of endogenous SAMHD1 expression in different cancer types [10,11,13,18]. However, in the previous studies, the role of SAMHD1 in the genome maintenance was examined exclusively in a biological context using cells exposed to genotoxic reagents [10,19,20]. Considering these results in light of our present result, a novel role for SAMHD1 in R-loop regulation has now been demonstrated, which when dysregulated leads to endogenous

genome-destabilizing stress (S6 Fig). Moreover, we suggest that SAMHD1 and R-loop are novel diagnostic markers and targets for patient stratification in anti-cancer therapy.

In recent decade, many tumor-suppressor genes such as BRCA1, BRCA2 and BLM and downstream nucleases including XPF and MRE11 have been suggested as proteins that are important for R-loop tolerance and removal [37,38,43–45]. More of classical proteins known for their roles in replication fork protection and transcription-coupled DNA repair are revealed as R-loop regulatory factors [46]. However, to develop R-loops as molecular markers in cancer and therapeutics, the context-dependent-physiological role of the R-loop should be considered. Uncontrolled R-loop formation can process to DSBs or cause ssDNA exposure [35]. On the other hand, physiological R-loop formation followed by DSBs can lay the foundation for DNA repair mechanisms such as chromosomal rearrangement, homologous recombination (HR), non-homologous end joining (NHEJ), or RNA-mediated HR [35]. Therefore, further investigation on the physiological meaning of the R-loop regarding to genome instability is needed.

Although TRC-dependent R-loops have been defined as "harmful R-loops", which promote cellular DNA damage responses [23], the understanding of other characteristics of R-loops in disease pathologies is still very limited. Even in fibroblasts of patients with AGS, AGS-specific DRIP peaks occupy in various regions other than replication origin adjacent gene bodies [18,25]. For example, mutations in RNASEH2A and RNASEH2B genes increased R-loops in the intergenic region and LINE. Additionally, AGS-associated gene mutations increase R-loops in LTRs. One could question that how these AGS unique R-loops can have significant impact on disease pathology as they only consist a small portion of total R-loops in the whole genome (S1A Fig). However, the AGS-unique R-loops have a higher number of DRIP peaks and cover much longer genomic regions than those of the healthy control-specific R-loops [25]. Therefore, the these gene mutations are certainly "burdens" to cells due to the high load of R-loop in the genome. It will be of great interest to investigate how these R-loops with diverse characteristics function in human diseases.

Since SAMHD1 has no binding affinity and cannot directly degrade DNA:RNA hybrids in vitro [16,47], its nucleic acid binding ability might be required for the recruitment of R-loop resolving factors to R-loops adjacent to replication origins. A possible mechanism of the R-loop regulatory function of SAMHD1 is that it might cooperate with R-loop resolving cofactors, such as helicases or nucleases. For example, SAMHD1 recruits and activates the MRE11 exonuclease at stalled replication forks [19]. Recently, a new function of the MRE11-RAD50-NBS1 complex in DNA replication-associated R-loop resolution has been reported [38]. In future investigations, it is possible to evaluate the collaborative activity of SAMHD1 together with known factors in R-loop regulation. Moreover, we identified two colorectal cancer-relevant SAMHD1 mutants, which cannot restore the increased TRCs in SAMHD1-depleted cells. In addition to SAMHD1 expression defects, SAMHD1 alternations may also be involved in R-loop-mediated genome instability, and cancer development may be due to of its inability to interact with R-loop regulatory cofactors. Future studies are expected to better characterize the physiological and pathological R-loops and SAMHD1-deficiency derived R-loops in cancer development.

## Materials and methods

### Experimental design

Sample size for immunofluorescence microscopy and comet assay in this study were chosen according to published guidelines and to our laboratory manual. All experiments were reproduced and the number of experimental replications for each experiment is stated in

figure legend. There was no data exclusion. No method of randomization or blinding was applicable.

## Cell culture

U2OS cells were cultured in Dulbecco's modified Eagle's medium (HyClone) supplemented with 10% (v/v) fetal bovine serum (FBS, HyClone), antibiotics mixture (100 units/ml penicillin-streptomycin, Gibco), and 1% (v/v) GlutaMAX-I (Gibco). Cells were incubated at 37°C under a 5% $CO_2$ atmosphere.

## Immunoblotting

Cells were lysed using RIPA buffer (50 mM Tris [pH 7.4], 150 mM sodium chloride, 0.5% sodium deoxycholate, 0.1% SDS, and 1.0% NP-40) supplemented with 10 μM leupeptin (Sigma-Aldrich) and 1 mM phenylmethanesulfonyl fluoride (Sigma-Aldrich) and boiled at 98°C for 10 min with SDS sample buffer prior to SDS-PAGE. Primary antibodies used were mouse monoclonal anti-SAMHD1 (ORIGENE, TA501956), monoclonal mouse anti-CHK1 (Cell Signaling, 2G1D5), monoclonal rabbit anti-phospho-CHK1 (Cell Signaling, 133D3), monoclonal rabbit anti-CHK2 (Cell Signaling, 6334T), monoclonal rabbit anti-phospho-CHK2 (Cell Signaling, C13C1), rabbit anti-GAPDH (AbFroontier), rabbit anti-mCherry (abcam, ab167453), and rabbit anti- HA-Tag (C29F4) (Cell Signaling, 3724). All primary antibodies were used at a dilution of 1:1000 for Western blotting. As secondary antibodies, peroxidase-conjugated anti-mouse IgG (115-035-062) and anti-rabbit IgG (111-035-003; Jackson Laboratories) were used at a 1:5000 dilution. Signals were detected using a SuperSignal West Pico chemiluminescence kit (Thermo Fisher Scientific).

## Plasmids

When indicated, cells were transiently transfected with the following plasmids. pICE-NLS-mCherry, pICE-RNaseH1-WT-NLS-mCherry, and pICE-RNaseH1-D10R-E48R-NLS-mCherrry (hereafter, Mut-RNaseH1-mCherrry) were purchased from Addgene (plasmid #60364, #60365, #60367). pcDNA3.1(+)-HA was purchased from Invitrogen (catalog #V79020), and pcDNA3.1(+)-WT SAMHD1-HA was prepared using restriction enzyme cloning. HB-GFP was prepared by cloning RNaseH1 HB domain into N3-EGFP (Clontech, catalog #6080–1) as previously described [37]. EGFP-tagged SAMHD1 was prepared by cloning full-length SAMHD1 into N3-EGFP.

## In vitro cell proliferation assay

SAMHD1-depleted (shSAMHD1) and control (shLuciferase) U2OS cells ($4 \times 10^3$ cells) were seeded into the wells of 24-well culture plates on day 1 of the experiment. The cells were then counted on days 3, 5, 7, 8, 10, 12, 14, and 15.

## Cell synchronization by double-thymidine block

For double-thymidine block of cell cycle synchronization, U2OS cells were seeded ($1.5 \times 10^5$ cells) into the wells of a 6-well plate on day 1. On day 2, 2.5 mM thymidine (Sigma, T1895) was added to the culture medium for the first thymidine block. After 18 h, the thymidine-containing culture medium was replaced with fresh growth medium without thymidine, and the cells were left for 9 h. The second round of thymidine block was given by adding 2.5 mM thymidine to the culture medium and incubating for another 18 h. If indicated, G1/S-phase–synchronized cells were released for 4 h to obtain an S-phase–synchronized cell population. The cells

were then washed with PBS twice and left in culture medium. Alternatively, the cells were harvested at indicated time points from 0 to 10 h post-release for further analysis.

## RNA interference and transfection

shRNAs targeting two different regions of the SAMHD1 3'-UTR and luciferase as a control were cloned into the pSUPER-retro-puro retroviral vector (laboratory collection). For the generation of SAMHD1-knockdown U2OS cells, cells were transfected with each shRNA cloned pSUPER-retro-puro retroviral vector using Lipofectamine 3000 reagent according to the manufacturer's instructions. The transfected cells were selected in 1 μg/mL puromycin for 1 week. The shRNA-targeting sequences of the SAMHD1 3'-UTR were as follows: 5'-GCATGCTGA AGCTAAGTAACT-3' and 5'-GGTACAAATTGGAACTAGAAA-3'. Lipofectamine 3000 (Invitrogen) transfection reagent was used for all plasmids, according to the manufacturer's protocol.

For target knockdown using siRNAs, the following synthetic duplex siRNAs were purchased from Dharmacon (siControl, ON-TARGET plus non-targeting siRNA (D001810-01, Dharmacon); siTHOC1, ON-TARGET plus Human THOC1 siRNA—SMART pool (L-019911-00, Dharmacon); siSETX, ON-TARGET plus Human SETX siRNA—SMART pool (L-021420-00, Dharmacon); siSAMHD1, siGENOME Human SAMHD1 siRNA–Set of 4 (MQ-013950-00-0020, Dharmacon); and siMRE11, ON-TARGETplus Human MRE11 siRNA-S-MART pool (L-009271-00-0005). U2OS cells were transfected with siRNA (10 nM) using the Lipofectamine 3000 reagent according to the manufacturer's instructions. Transfections with ON-TARGETplus Non-targeting pool (Dharmacon) were performed in parallel as a negative control.

## Immunofluorescence microscopy

For the immunofluorescence assays of S9.6 nuclear signals, SAMHD1-depleted or control U2OS cells were fixed with 100% ice-cold methanol for 10 min on ice and then incubated with 100% ice-cold acetone for 1 min. The slides were washed three times with 1× PBS and then incubated with or without 60 U/mL of RNase H (M0297S, NEB) at 37°C for 36 h or left untreated. The slides were subsequently briefly rinsed three times with 2% BSA/0.05% Tween (in PBS) and incubated with mouse anti-DNA:RNA hybrid S9.6 (Kerafast, ENH001) used at a dilution of 1:100 and rabbit anti-nucleolin (Abcam, ab22758) at a dilution of 1:300 in 2% BSA/0.05% Tween (in PBS) for 4 h at 4°C. The slides were then washed three times with 2% BSA/0.05% Tween (in PBS) and incubated with goat anti-rabbit AlexaFluor-488–conjugated (Invitrogen, A-11008) and goat anti-mouse AlexaFluor-568–conjugated (Molecular Probes, A11004) secondary antibodies (1:200) for 2 h at RT. The slides were then washed three times with 2% BSA/0.05% Tween (in PBS) and mounted using ProLong Gold AntiFade reagent (Invitrogen). Images were obtained using an inverted microscope Nikon Eclipse Ti2, equipped with a 1.45 numerical aperture (NA) Plan apo λ 100x oil objective and an sCMOS camera (Photometrics prime 95 B 25mm). For each field of view, images were taken with DAPI395, GFP488, and Alexa594 channels using NIS-Elements software. For quantification analysis, binary masks of nuclei and nucleoli were generated using the ROI manager and auto local thresholding by the Phansalker method in ImageJ [48]. Centers of R-loop puncta were detected by TrackNTrace [49]. The intensity of R-loop puncta was quantified by adding the intensity of 9 pixels from $3 \times 3$ squares. R-loop puncta inside the nucleus and outside the nucleoli were analyzed using custom code written in MATLAB. For the immunofluorescence assays of H2AX phosphorylation, cells were fixed with 3.7% formaldehyde in PBS. The slides were then incubated with γH2AX rabbit (Cell Signaling, 20E3), phospho-ATR (Ser428) rabbit

(Cell Signaling, 2853S), and phospho-ATM (S1981) rabbit (Abcam, ab81292) antibodies, as wll as goat anti-HA-Tag (Genescript, A00168), at 1:200 dilution with 2% BSA (in PBS) for 2 h at RT. The slides were washed three times with 2% BSA/0.05% Tween (in PBS) and incubated with goat anti-rabbit AlexaFluor-568–conjugated (Molecular Probes, A11011), goat anti-rabbit AlexaFluor-488–conjugated (Invitrogen, A-11008), and 1:200 rabbit anti-goat-FITC-conjugated (Jackson lab, 305-095-047) antibodies at 1:200 dilution for 1 h at RT. Subsequently, the slides were washed three times with 2% BSA (in PBS) and then mounted using ProLong Gold AntiFade reagent (Invitrogen). To obtain images, the mounted specimens were visually scanned, and representative images were acquired using a Zeiss LSM 710 laser scanning confocal microscope (Carl Zeiss).

## Comet assay

SAMHD1-depleted (shSAMHD1) and control (shLuciferase) U2OS cells ($1.5 \times 10^5$ cells) were seeded 1 day prior to plasmid transfection. Cells were transiently transfected with either full-length SAMHD1-HA expression vector or empty vector using Lipofectamine 2000 according to the manufacturer's instructions. For RNase H1 and SAMHD1-HA reconstitution experiments, cells were transfected with either protein expression vector or empty vector. At 48 hours post-transfection, an alkaline assay was performed on U2OS cells using an OxiSelect Comet Assay kit (CELL BIOLABS, STA-351) according to the manufacturer's instructions. Fifty comets per slide were scored using CometScore software (TriTek).

## DNA-RNA hybrid immunoprecipitation (DRIP)

DRIP was performed as described previously in [50] with the following modifications. In brief, corresponding cells were harvested and genomic DNA was extracted with DNA purification kit (QIAGEN, 51106) by manufacturer's instruction. 20ug of extracted nucleic acids were fragmented using a restriction enzyme cocktail (20U each of EcoRI, BamHI, HindIII, BsrBI and Xho I; New England Biolabs) overnight at 37˚C. Half of the fragmented nucleic acids were digested with 40U RNase H (New England Biolabs) to serve as negative control, overnight at 37˚C. The digested nucleic acids were cleaned with standard phenol-chloroform extraction and re-suspended in DNase/RNase-free water. DNA-RNA hybrids were immunoprecipitated from total nucleic acids using 10ug of the mouse anti-DNA-RNA hybrid S9.6 (Kerafast, ENH001) in binding buffer (10 mM NaPO4 pH 7.0, 140 mM NaCl, 0.05% Triton X-100) and incubated overnight at 4˚C. Dynabeads Protein A (Invitrogen, 10001D) were used to pull-down the DNA-antibody complexes, by incubating 3hr at 4˚C. Isolated complexes were washed twice with binding buffer and once with TE buffer for 5min at RT, before elution with elution buffer (50 mM Tris pH8.0, 10mM EDTA, 0.5% SDS, 2.5ug protease k for 30min at 55˚C. Subsequently, DNA was purified following the standard phnol-chloroform extract method and used for qPCR.

## Quantitative real-time PCR (qPCR)

Equivalent amounts of purified genomic DNA (50 $\sim$ 100 ng) from each sample were analyzed by qPCR. For RT (reverse transcription)-qPCR, 1 $\sim$ 2 µg of RNA was reverse-transcribed using the ReverTra Ace qPCR RT Kit (TOYOBO) following the manufacturer's instructions. The resulting complementary DNA (cDNA) was diluted with sterile deionized H2O (1:6). RT-qPCR reactions were performed in TOPreal qPCR PreMIX (Enzynomics) with 2 µl of the diluted cDNA. The total reaction volume was 20 µl, and all reactions were performed in triplicate. The PCR reactions were performed using the iCycler iQ real-time PCR detection system (Bio-Rad). Data were normalized according to the expression levels of

β-actin or MDM2, as indicated. qPCR analyses were performed using the specific primers listed in S1 Table.

## Proximity ligation assay (PLA)

For the PCNA–RNAPII PLA, cells grown on coverslips were fixed with 3.7% formaldehyde for 20 min at room temperature and permeabilized with 0.2% Triton X-100 for 5 min. Cells were then blocked with 1X blocking solution (Merck, DUO92102) for 1 h at 37˚C in a humidity chamber. After blocking, cells were incubated with primary antibody overnight at 4˚C (for PCNA–RNAPII PLA; 1:500 rabbit anti-PCNA antibody (ab18197, Abcam), 1:100 mouse anti-RNAPII (Santa Cruz, sc-56767), and 1:200 goat anti-HA-Tag (Genescript, A00168). The next day, after washing with 1× Wash buffer A twice (Merck, DUO92102), cells were incubated with pre-mixed Duolink PLA plus, PLA minus probes, and 1:200 rabbit anti-goat-FITC-conjugated (Jackson lab, 305-095-047) antibodies for 1 h at 37˚C. The subsequent steps in the proximal ligation assay were carried out using the Duolink PLA Fluorescence kit (Sigma) according to the manufacturer's instructions. To obtain images, the mounted specimens were visually scanned, and representative images were acquired using a Zeiss LSM 710 laser scanning confocal microscope (Carl Zeiss). For quantification analysis, binary masks of nuclei were generated by consecutive steps of Gaussian blur, Bernsen auto local thresholding, watershed segmentation, and Analyze Particles function in ImageJ. PLA puncta were detected and analyzed by custom code written in ImageJ macro language utilizing Gaussian blur, manual thresholding, and Analyze Particles function.

## Cell-cycle analysis

Cell cycle analysis was performed by both PI staining and pulse-chase BrdU (Sigma) incorporation, as previously described [10]. For BrdU pulse-chase experiments, a pulse of 10 mM BrdU was added to the cell culture 30 min before harvesting. Afterwards, BrdU-free medium was added to the cell culture, and analyses were performed every 3 h for an additional 12 h. Cells were fixed with 70% ethanol (EtOH), and the DNA was denatured using 1 ml of 1 N HCl/0.05% Triton X-100 and then neutralized with 1 ml of 0.1 M $Na_2B_4O_7$. Cells were blocked using PBS/1%BSA/0.5% Tween-20 and incubated with mouse anti-BrdU primary antibody (Cell Signaling, 5292), followed by goat-donkey-mouse AlexaFluor-488–conjugated secondary antibody (Life Technologies, A21202). The cells were then subjected to PI staining followed by flow cytometry analysis. For PI staining, cells were harvested and fixed in pre-chilled 70% EtOH and then stored overnight at −20˚C. Fixed cells were recovered and resuspended in PBS containing 100 μg/ml of RNase A (Thermo Fisher Scientific, R1253) and 10 μg/ml of PI and incubated for 12 h at 4˚C and then analyzed using a Flow-Activated Cell Sorter Canto II (BD Bioscience) and Flowjo software (Flowjo).

## RNA/DNA hybrid dot blotting

Total genomic DNA was extracted using a QIAmp DNA mini kit (QIAZEN, 51304) according to the manufacturer's instructions. Genomic DNA (1.2 μg) was treated with 2 U of RNase H (NEB, M2097) per microgram of genomic DNA for 48 h at 37˚C, with half of the sample left untreated but denatured. Half of the DNA sample was probed with S9.6 antibody (1:1000), and the other half was probed with anti-ssDNA antibody (MAB3034, Millipore, 1:10000).

## Annexin V-FITC/ PI double-staining assay

Annexin V-FITC/PI double staining was performed using an Annexin V-FITC kit (BD Bioscience, 556547). The cells were labeled with FITC-conjugated Annexin V and 5 μl of PI according to the manufacturer's instructions. After incubation in the dark for 15 min at RT, the cells were analyzed by flow cytometry. The AnnexinV-FITC-negative/PI-negative population was regarded as normal, whereas the Annexin V-FITC–positive/PI–negative and Annexin V-FITC–positive/PI–positive populations were taken as measurements of early and late apoptotic/necrotic cells, respectively.

## HB-GFP retention fluorescence-activated cell sorting assay

SAMHD1-depleted (shSAMHD1) and control (shLuciferase) U2OS cells were transfected with HB-GFP. At 24 h post-transfection, cells were harvested and resuspended in ice-cold PBS containing 2% FBS and then permeabilized for 4 min with 0.05% Triton X-100 in the presence of protease inhibitor cocktail (Cell Signaling, 5872S). Three volumes of 100% ice-cold ethanol were added, and the sample was kept on ice for 1 h and then co-stained with PI and then analyzed via flow cytometry.

## Data collection and TCGA dataset analysis

RNA sequencing (RNA-seq) data and clinical information for patients with lung squamous cell carcinoma, lung adenocarcinoma, bladder urothelial carcinoma, COADREAD, breast invasive carcinoma, and prostate adenocarcinoma were downloaded from TCGA dataset using the UCSC Xena browser, as previously described [51]. Gene expression profiles were generated experimentally using an Illumina HiSeq 2000 RNA sequencing platform by the University of North Carolina TCGA Genome Characterization Center. RNA-seq data were quantified using RSEM, as described previously [52]. Associations between the SAMHD1 expression and clinicopathologic features of patients with each type of cancer were analyzed and plotted. Overall survival of COADREAD patients was determined using clinical and demographic information downloaded from the Pan-cancer Project TCGA dataset. The patients were separated into two groups depending on the expression levels of SAMHD1, TREX1, RNASEH2A, or RNASEH2B, with the cut-off being the optimal cut-point. Survival curves for high-expressing and low-expressing groups were produced using the Kaplan-Meier method, and log-rank tests were used to assess differences in survival between groups.

## Bioinformatics analysis

DRIP-seq data from control, AGS1, AGS2, AGS4 and AGS5 patients were obtained from the Gene Expression Omnibus database under the accession number GSE57353 [25]. Fastq files of each replicates were filtered to obtain files with 100nt read length and those identical lengths of reads were used for analysis. For each fastq files, adapter sequences were trimmed with cutAdapt, and artifacts were removed by fastx-toolkit. Trimmed reads were mapped to hg19 with bowtie2. Alignment was performed with '—very-sensitive' option & 1nt mismatch was allowed on each reads. Duplicates reads were removed by samtools. DRIP-signal for each 100nt region was defined as mapped read-count, which was normalized to 10M total duplicate-removed read-count. To analyze DRIP-seq read-count of AGS patients for each region, we obtained dataset from under the accession number GSE93368 and custom python scripts from GitHub (https://github.com/cimprichlab/Hamperl_et_al) [23]. The custom python script was modified according to the modified datasets as described above. The DRIP-seq data for control, AGS1, AGS2, AGS4, and AGS5 patients were aggregated to these classified origins,

and then DRIP-seq read-count profiles were calculated over a 24 kb window around origins or centers of gene bodies. As previously described [23], in order for HO collision to have a negative genome coordinate from origins, the windows were inverted in prior to aggregation in gene bodies those are transcribed on negative strands. The modified custom python scripts used to determine DRIP-seq signal across TRC regions were made available under GitHub (https://github.com/spiritmage72/samhd1_rloop). Metaplot (Fig 1A) and boxplot (Figs 1B, S1B and S1C) were obtained with means or medians, which were bootstrapped 10,000 times.

Raw fastq files were imported from published data under accession number GSE57353. From the 484 genes selected for DRIP-seq analysis, we investigated 467 genes whose ENSEMBL ID matched the expression level dataset. To identify DEGs of control versus those of AGS patient fibroblasts, we used the DESeq2 tool as previously described [53]. The detection criteria to identify DEGs were |log2(fold change)| >2 and adjusted P value <0.01.

## Statistical analyses

Statistical parameters, including the number of biological replicates (n), standard deviation, and statistical significance, are indicated in each figure and figure legend. Statistical analysis was performed with GraphPad Prism 5. Statistical significance was determined by two-tailed Student's t test, two-way ANOVA or two-tailed Mann-Whitney U test, as indicated.

For DRIP-seq data bioinformatics analyses, statistical analysis was performed with python package, SciPy. For DRIP-seq read-count enrichment comparison, a contingency table was computed for each AGS patient fibroblast DRIP-seq sample by normalized read-counts mapped to 12 kb or 6 kb window. The adjusted P values were calculated by using the Fisher's exact test followed by Bonferroni correction for multiple test. Means or medians used for statistical analysis were not bootstrapped.

## Supporting information

**S1 Fig. R-loop accumulation at the origins of gene bodies in SAMHD1-deficient AGS patient fibroblasts. A,** Average mapped DRIP-seq read accumulation around random genomic position (n = 50,000) in fibroblasts from control (yellow) and AGS patients with the indicated gene mutations (blue, SAMHD1; pink, TREX1; green, RNASEH2B; purple, RNASEH2A). For individual graph, the line indicates the mean DRIP-seq read-count of each genomic region of 24 kb windows centered on random genomic position. **B,** Volcano plot of RNA-seq transcriptome data displaying the pattern of gene expression values for fibroblasts from control and AGS patients (AGS5, AGS1, AGS2, and AGS4). Significantly differentially expressed genes (DEGs) (|log2(fold change)| >2 and adjusted P value <0.01) are highlighted in red (upregulated) or blue (downregulated). None of the DEGs are highlighted in black. Selected genes used for DRIP-seq analysis (n = 467) are highlighted in dark green. The number of DEGs among selected genes used for DRIP-seq analysis for each AGS patient are indicated. **C,** Box graph indicating quantified average DRIP-seq read-count across the 6 kb window upstream from replication origins of the gene bodies (n = 727), where HO collision sub-regions exist. **D,** Box graph indicating quantified average DRIP-seq read-count across a 6-kb window downstream from replication origins of the gene bodies (n = 727), where CD collision sub-regions exist. In **C** and **D**, the orange line at the center of the boxes indicates the median, and the boxes indicate 1st and 3rd quartiles. Statistical significance was assessed using Fisher's exact test followed by Bonferroni correction.
(TIF)

**S2 Fig. Depletion of SAMHD1 does not affect cellular apoptosis or cell cycle arrest. A,**
SAMHD1 in U2OS cells transfected with vectors expressing shRNA targeting luciferase control (shLuc) and two different SAMHD1 UTRs (shSAMHD1-#1 and shSAMHD1-#2). **B,** In vitro cell proliferation assay of SAMHD1-depleted or control U2OS cells. Data represent the mean ± SD, n = 3. The P value was calculated according to two-way ANOVA. **C,** Representative flow plot of Annexin V-FITC assay of SAMHD1-depleted and control U2OS cells. Apoptosis was determined by staining cells with FITC-conjugated Annexin V and PI, followed by flow cytometry analysis. Four populations are indicated as Q1, necrotic; Q2, late apoptosis; Q3, live; and Q4, early apoptotic. **D,** Quantification of Annexin V-FITC apoptosis assay results showing the percentage of cell death modes: live cells, early apoptosis, and late apoptosis/ necrosis in SAMHD1-depleted and control U2OS cells. Statistical significance was assessed by two-way ANOVA. Data represent the mean ± SEM (n = 3, P < 0.001). **E,** Representative cell cycle profile analysis by flow cytometry following labeling of cells with BrdU. SAMHD1-depleted and control U2OS cells were pulsed with BrdU. The BrdU-positive population was followed over time as it transitioned through the cell cycle.
(TIF)

**S3 Fig. Depletion of SAMHD1 leads to increased DNA damage. A,** Summary of experimental design for Fig 2C, representing experimental process of double thymidine block followed by time course sample harvest. **B,** Representative images of comet assays performed under alkaline electrophoresis conditions in SAMHD1-depleted and control U2OS cells, in the absence or presence of ectopic SAMHD1-HA expression. **C,** Quantitative analysis of comet tail moment lengths for each condition described in **B**. Statistical significance was assessed using two-tailed Student's *t*-test (n >50). **D,** Summary of experimental design for Fig 2F, representing experimental process to obtain heterogeneous (Asyn) or S phase-synchronized cellular population by double thymidine block. **E,** Summary of experimental design for **F,** representing experimental process of SAMHD1-HA reconstitution followed by double thymidine block to obtain S phase-synchronized cellular population. **F,** Immunoblots of CHK1 phosphorylated on S345 (p-CHK1), CHK2 phosphorylated on T68 (p-CHK2), CHk1 and CHk2 in S phase-synchronized SAMHD1-depleted and control U2OS cells, in absence or presence of ectopic SAMHD1-HA expression in a does dependent manner (n = 3). **G,** Representative images of comet assays performed under neutral conditions in SAMHD1-depleted and control U2OS cells. **H,** Quantitative analysis of comet tail moment lengths for each condition described in **G**. Statistical significance was assessed using two-tailed Student's *t*-test (n >50).
(TIF)

**S4 Fig. Accumulation of R-loops in SAMHD1-depleted cells. A,** Indicated gene expression as measured by RT-qPCR in control (siControl) or indicated siRNAs treated U2OS cells. Data represent mean ± SEM, n = 3, P values were calculated according to two-tailed Student's *t*-test. **B,** Schematic representation of HB-GFP retention assay. **C,** Quantification analysis of HB-GFP retention assay co-stained with PI. Data represent mean ± SEM percentages of GFP positive cell population. Statistical significance was assessed using two-tailed Student's *t*-test (n = 3). P > 0.05; n.s, not significant. **D,** Quantification analysis of HB-GFP retention assay co-stained with PI. Data represent mean ± SEM percentages of S phase cell population. Statistical significance was assessed using two-tailed Student's *t*-test (n = 3). P > 0.05; n.s, not significant. **E,** Immunoblots of SAMHD1 in cells described in Fig 3G, showing ectopic expression of GFP-tagged SAMHD1 in SAMHD1-depleted and control U2OS cells. **F,** Dot blot analysis of R-loops in genomic DNA extracted from SAMHD1-depleted and control U2OS cells (n = 3). The total cellular genomic DNA from each cell line was incubated with or without RNase H in prior to loading for blotting, as indicated. The genomic DNA loaded membrane were

processed to immunoblotting with anti-S9.6 and anti-ssDNA antibodies to verify equivalent DNA loading. Fold-induction was normalized to the control U2OS cells by quantifying the dot intensity of the blots and calculating the ratios of S9.6 signal to total amount of genomic DNA. **G and H,** Indicated gene expression as measured by RT-qPCR in control or SAMHD1-depleted U2OS cells. Data represent mean ± SEM, n = 3, P values were calculated according to two-tailed Student's *t*-test. P > 0.05; n.s, not significant. **I,** DRIP-qPCR using the anti S9.6 antibody at GBE1 HO, GBE1 CD, MAP4K4 HO, MAP4K4 CD, CALD1 HO, CALD1 CD, PALM2-AKAP2 HO and PALM2-AKAP2 CD are shown in SAMHD1-depleted and control U2OS cells. Pre-immunoprecipitated samples were untreated (-) or treated (+) with RNase H, as indicated. Values are relative to those of control U2OS cells. Data represent mean ± SEM the statistical significance was assessed using the one-way ANOVA (n = 2).
(TIF)

**S5 Fig. Colorectal cancer-relevant SAMHD1 mutants are associated with TRCs in U2OS cells. A,** Immunoblots of SAMHD1 in U2OS cells transfected with vector expressing indicated SAMHD1 mutants tagged with HA. **B,** Genomic DNA extracted from U2OS cells described in **A** were analyzed by quantitative PCR. The relative DNA copy number for each vector was determined by Puromycin resistance sequence and normalized to MDM2. Data represent mean ± SEM, n = 2, P values were calculated according to two-tailed Student's *t*-test. P > 0.05; n.s, not significant. **C,** Representative images of proximity ligation assay (PLA) between PCNA and RNA polymerase II (RNAPII) in S phase synchronized SAMHD1-depleted and control U2OS cells, in the absence or presence of indicated SAMHD1 mutant ectopic expression (SAMHD1-HA) (n = 2). Cells were subjected to PLA (red), and co-stained with HA (green) and DAPI (blue). **D,** Quantification analysis of number of PLA foci per nucleus in **C**. The mean value for each data is shown as a red line. Statistical significance was assessed using the two-tailed Mann-Whitney *U* test (n > 70).
(TIF)

**S6 Fig. Model of the role of SAMHD1 at transcription-replication conflict-derived R-loops.** In normal cells (left), R-loops formed by transcription-replication collision, are resolved by SAMHD1, which allows successful DNA replication and RNA transcription. In SAMHD1-deficient cells (right), unresolved R-loops at transcription-replication conflict regions showed genomic instability.
(TIF)

**S1 Table. Oligonucleotides used in this study.**
(DOCX)

**S1 Data. Source Data: Spreadsheet of source data shown in this study.**
(XLSX)

## Acknowledgments

We are grateful to Dr. Dong-Yeon Cho for help with bioinformatics analysis and discussion.

## Author Contributions

**Conceptualization:** Kiwon Park, Jeongmin Ryoo, Heena Jeong, Kwangseog Ahn.

**Data curation:** Kiwon Park, Jeongmin Ryoo, Minsu Kim, Sungwon Lee, Sung-Yeon Hwang, Jiyoung Ahn, Doyeon Kim, Hyungseok C. Moon, Daehyun Baek, Kwangsoo Kim, Hye Yoon Park, Kwangseog Ahn.

**Formal analysis:** Minsu Kim, Jiyoung Ahn, Doyeon Kim, Hyungseok C. Moon, Daehyun Baek, Kwangsoo Kim, Hye Yoon Park.

**Funding acquisition:** Kwangseog Ahn.

**Investigation:** Kiwon Park, Jeongmin Ryoo, Heena Jeong, Sungwon Lee, Sung-Yeon Hwang, Kwangseog Ahn.

**Methodology:** Kiwon Park, Jeongmin Ryoo, Heena Jeong, Sungwon Lee, Sung-Yeon Hwang, Kwangseog Ahn.

**Project administration:** Kiwon Park, Jeongmin Ryoo, Kwangseog Ahn.

**Software:** Minsu Kim, Jiyoung Ahn, Doyeon Kim, Hyungseok C. Moon, Daehyun Baek, Kwangsoo Kim.

**Supervision:** Kwangseog Ahn.

**Validation:** Kiwon Park.

**Visualization:** Kiwon Park.

**Writing – original draft:** Kiwon Park, Jeongmin Ryoo.

**Writing – review & editing:** Kiwon Park, Jeongmin Ryoo, Heena Jeong, Minsu Kim, Sungwon Lee, Sung-Yeon Hwang, Kwangseog Ahn.

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
