## [Decision Letter · Decision Letter 0]

10 Nov 2020

Dear Dr Ahn,

Thank you very much for submitting your Research Article entitled 'Aicardi-Goutières syndrome-associated gene SAMHD1 preserves genome integrity by preventing R-loop formation at transcription-replication conflict regions' to PLOS Genetics. Your manuscript was fully evaluated at the editorial level and by independent peer reviewers. The reviewers appreciated the attention to an important problem, but raised some substantial concerns about the current manuscript. Based on the reviews, we will not be able to accept this version of the manuscript, but we would be willing to review again a much-revised version. We cannot, of course, promise publication at that time. In particular, we would like the see more molecular evidence underlying the links between SAMHD1 function and R-loop biology to support the model presented in this paper. Furthermore, all reviewers have raised concerns regarding quality of the data and lack of appropriate controls. This includes lack of gene specific validations and correlation between different cell lines used in the paper, all of which need to be addressed in the revised version.

If you decide to revise the manuscript for further consideration at PLOS Genetics, please aim to resubmit within the next 60 days, unless it will take extra time to address the concerns of the reviewers, in which case we would appreciate an expected resubmission date by email to plosgenetics@plos.org.

[LINK]

We are sorry that we cannot be more positive about your manuscript at this stage. Please do not hesitate to contact us if you have any concerns or questions.

Yours sincerely,

Natalia Gromak, PhD

Guest Editor

PLOS Genetics

Peter McKinnon

Section Editor: Cancer Genetics

PLOS Genetics

Reviewer's Responses to Questions

**Comments to the Authors:**

Reviewer #1: The manuscript “Aicardi-Goutières syndrome-associated gene SAMHD1 preserves genome integrity by preventing R-loop formation at transcription-replication conflict regions” by Park et al. reports on R-loop increases in fibroblasts of human AGS patients with mutations of distinct genes. Then, bioinformatic studies provide some evidence that SAMHD1 deficiency may delay replication by increasing R-loops near replication origins located in the body of genes and activation of checkpoint signalling pathways. Finally, bioinformatic analyses of TCGA datasets correlate SAMHD1 expression with cancer progression.

The data are novel, correctly presented and discussed. However, some critical issues are present in the paper that need to be addressed:

- DRIPseq data: it would be interesting to present and discuss the overall distribution of R-loops in the genome of fibroblasts bearing different mutated AGS genes. Is the overall distribution changed with specific gene mutations ? In other words: the changes reported in Fig 1 are unique or other changes are present elsewhere ?

- AGS gene mutations might have an effect on gene transcription and activated ORI may be different among cell types. Thus, some validations of the DRIPseq results should be provided at single-gene levels. At selected genes, authors should check if their expression levels, ORI activity and R-loop levels are altered (or not), and discuss the results with respect to DRIPseq findings.

- Cellular data of Fig 2 are not fully convincing. Panel A reports a difference from 3.5% to 1.7%. Can that be an experimental fluctuation ? Is such a difference biological relevant ? Moreover, double thymine block is known to be very stressful for cells, inducing DNA damage response (also shown in panel E in fig 2). At least, a better discussion of these points should be included in the text.

- Panel G, Figure 2: are images mis-labelled ?

- Activation of DNA damage response should be demonstrated by checking phosphorylation of H2AX, and both ATR and ATM kinases. This is particular interesting as the authors show that R-loops are increased in the half of ORI where replication/transcription head-on collision may occur.

- Immunofluorescence microscopy images are very poor. Looking the images, I am not convincing that R-loops increase in the nucleus of SAMHD1-deficient cells. I know that immunofluorescence microscopy with S9.6 may be tricky, however, I would suggest not to increase the fluorescence intensity of images to avoid to decrease signal/background ration. In addition, RNaseH1 treatment samples should be shown along with controls in which the same treatment was performed but without RNaseH1 addition in the buffer.

- With respect to the previous point, how may an overall R-loop increase (as seen in immunofluorescence images) compare with a rather selective and modest increase at ORIs in transcribed genes ?

- I suggest to edit the text for English syntax and fluency.

Reviewer #2: In this manuscript, Park, Ryoo and colleagues report the observation that SAMHD1, a factor mutated in the Aicardi Goutières syndrome, prevents the accumulation of R-loops and protect human cells from head-on transcription-replication conflicts and associated genomic instability. This view is largely based on bioinformatic analyses of published datasets showing that SAMHD1 deficiency, but not related AGS mutations, is associated with increased R-loop levels at a subset of genes containing replication origins. In addition, they provide evidence that SAMHD1-depleted U2OS cells have a delayed S phase accumulate spontaneous DNA damage and activate DNA damage response pathways. Finally, they show that colon cancer patients that have a low expression of either SAMHD1 or RNase H2A have a poor survival. On the basis of these data, the authors propose that SAMHD1 contributes to the resolution of R-loops and prevents genomic instability in cancer cells. Although this model is attractive, the data presented in the manuscript are not sufficient to support it and mechanistic insights in how SAMHD1 could execute this important function is missing. Further work is therefore required to clearly establish that SAMHD1 plays a role in R-loop metabolism.

Major issues:

1. In Figure 1, the authors reproduced the analysis performed by the Cimprich lab (Hamperl, Cell 2017) to compare the distribution of R-loops in genes containing a replication origin in the gene body versus genes that do not contain replication origins. To this end, they used DRIP-seq data generated by the Chedin lab (Lim, ELife 2015) to position R-loops in primary fibroblasts from AGS patients and OK-seq data generated by the Hyrien lab to determine fork direction (Petryk, Nat Commun 2016). This analysis revealed a striking difference in the distribution of R-loops at origin-containing genes in SAMHD1-deficient cells, but not in other AGS mutants. Although this result is intriguing, it is not sufficient to conclude that SAMHD1 resolves R-loops at genes experiencing transcription-replication conflicts. Indeed, it is well established that the position of replication origins varies between cell lines. The Cimprich lab mapped R-loops in HeLa cells and analyzed their distribution relative to forks using OK-seq data also generated in HeLa cells. Here, the authors compared R-loops mapped by the Chedin lab in primary fibroblasts relative to forks mapped in HeLa cells, which is like comparing apples and pears. Moreover, SAMHD1-deficient cells have altered dNTP pools and slower replication forks, which may also impact on the replication program and explain the difference with other AGS mutants. In conclusion, although the difference observed between SAMHD1-deficient fibroblasts and other AGS mutants is clear, this analysis needs to be performed in the same cell lines, or at least confirmed for a subset of loci by other methods.

2. The authors claim that they report for the first time a role for SAMHD1 in S-phase progression in unchallenged cells but this statement is incorrect. Other groups have already reported that cell cycle progression is delayed and forks are slower in the absence of SAMHD1 (see for instance Franzolin, PNAS 2013; Kretschmer, Annals of the Rheumatic Diseases 2015; Coquel, Nature 2018). This statement should therefore be toned down. Along the same line, the authors repeatedly state that SAMHD1 is the only AGS-associated gene involved in cancer. RNase H2B has been reported as a colorectal tumor suppressor (Aden et al. https://doi.org/10.1053/j.gastro.2018.09.047). RNase H2A is involved in human gliomagenesis through the regulation of cell proliferation and apoptosis (Dai et al. https://doi.org/10.3892/or.2016.4802).

3. The data showing that SAMHD1-depleted cells have increased R-loop levels (Fig. 3) are not really convincing. The reliability of immunofluorescence to quantify R-loops is questioned in the field and the flow cytometry approach would require positive controls to position the gate. The slot blot is not quantified, lacks positive controls and the number of biological replicates is not indicated. DRIP-qPCR experiments on a set of genes should be performed.

4. From the data shown in Fig.1, one would predict that the overall amount of R-loops in SAMHD1-deficient cells should not dramatically change if the effect is restricted to genes containing active origins. This issue needs to be clarified textually.

5. In Fig. 4, the authors convincingly show that the overexpression of RNase H1 reduces the formation of spontaneous DSBs in SAMHD1-depleted cells but they would have made their case stronger by showing that these breaks occur during S phase. Moreover, they need to show that the slower S phase and increased gamma-H2AX levels observed in the absence of SAMHD1 (Fig. 2) is also rescued by RNase H1. The authors could also assess fork progression in SAMHD1-depleted cells and test whether fork progression defects can be suppressed by transcription inhibition or RNase H1 overexpression.

6. The association of cancer patient survival to SAMHD1 level of expression is interesting, but can it be generalized to other types of cancers? Do tumors expressing low levels of SAMHD1 and RNase H2A constitute different groups? It has been reported that SAMHD1 levels impact on the response to chemotherapy, notably when nucleoside analogues (ara-C) are used (Herold, Nat Med 2017). Is that relevant to colon adenocarcinoma treatment?

Minor issues:

1. Fig. S1C: the p-value for DRIP-seq enrichment at CD collision is labelled as P=0.0707, however in the text, the authors indicate P=0.707 (line 158). Which value is correct?

2. Fig. 2G: Control cells show a very high gamma-H2AX signal. The labeling is incorrect.

3. Fig. 4C: The blot for p-CHK2 should be improved.

Reviewer #3: In the manuscript entitled “Aicardi-Goutières syndrome-associated gene SAMHD1 preserves genome integrity by preventing R-loop formation at transcription-replication conflict regions”, the authors show first in a bioinformatic analysis that R-loop levels are increased at particular gene body regions prone to Head-ON transcription-replication conflicts in fibroblasts mutated in the dNTPase and phosphorolytic 3ʹ- 5ʹ -exoribonuclease SAMHD1 compared to control cells (Figure 1)

Depletion of SAMHD1 in U-2OS cells leads to replication stress and DNA damage in S-Phase (Figure 2) which correlates with cellular R-loop accumulation (Figure 3). Removal of R-loops can rescue these effects (Figure 4) suggesting a more direct link between R-loop accumulation and the genomic instability phenotypes of SAMHD1 depleted cells. Finally, TCGA database analysis reveals a correlation between SAMHD1 expression level and patient survival rate (Figure 5).

Overall, I think the manuscript addresses an interesting question relevant to a broad audience. The data are generally of high quality and convincing. However, I do have a few major concerns that need to be addressed and solidified prior to publication in Plos Genetics.

Major Points:

1) The bioinformatic analysis was adapted from a previous study showing that genomic regions prone to HO conflicts have higher R-loop levels than CD (Hamperl et al., Cell 2017). This study used OK-Seq data in HeLa cells to determine replication fork directionality. As I could not find specific information in the material and method section, did the authors use the same data set (derived from HeLa) to determine the locations of origins for the analysis in the AGS mutant cell lines? As usage of replication origins is stochastic and may vary to a significant extent between different cell types and genetic backgrounds, it is unclear to me whether one can confidently conclude that these origins mapped in HeLa cells also exist in the fibroblast cell lines. It would be best if OK-Seq data are available in the exact same cell lines. At the minimum, the authors could analyze additional OK-Seq data sets from other human cell lines and then analyze if and to what extent these origins are conserved between multiple cell lines

2) Another point regarding this analysis in Figure 1. Have the authors looked into gene expression levels of these 500 genes whether they show significant levels of transcription in the different mutants versus control cells?

3) Based on the phenotypes observed in U-2OS cells (Figures 2-4), the authors have clear indications that SAMHD1 knockdown leads to DNA damage-inducing accumulation of RNA:DNA hybrids on a global/cellular level but there is no effect in the overall level of R-loops in SAMHD1 mutant cells compared to control cells (Figure S1A). Is this an effect of knocking down the protein (in U-2OS cells) versus expressing a mutant version of the enzyme in the fibroblast that can induce a dominant negative effect? I think it would be important if the authors could confirm their findings in U-2OS cells by introducing the exact same mutation of SAMHD1 in U-2OS cells in order to make the two different experimental systems/cell lines more comparable. A major selling point of the paper is the relevance of SAMHD1 towards many types of cancer but I believe this point is only proven if the same effects of R-loops at sites of transcription-replication conflicts can be shown with the cancer relevant mutated version of the enzyme.

4) The DNA damage accumulation in SAMHD1-depleted cells leading to Chk1, Chk2 and yH2AX activation shows nicely an S-phase specific effect of SAMHD1 (Figure 2). The alkaline comet assay shows all kinds of DNA damage including ssDNA and DSBs. Could the authors also perform a neutral comet assay to get a better understanding what type of damage is inflicted upon SAMHD1 loss?

5) Figure 3: Can the authors show that the accumulation of R-loops is specific to S-phase cells (maybe co-labelling with an EdU pulse)? Alternatively, co-stain with PI to investigate the DNA content of cells that have elevated HB-GFP signal (Figure 3D)?

6) A more conceptual point is that SAMHD1 was recently shown to associate with active replication forks, thereby promoting degradation of nascent DNA at stalled replication forks by stimulating the exonuclease activity of MRE11, thereby mitigating an autoimmune response (Coquel et al., Nature 2018). I guess one model could be that these sites where SAMHD1 acts on are replication forks stalled due to transcription-replication conflicts. Can the authors confirm this by using a more direct readout that shows elevated levels of conflicts in SAMHD1 knockdown conditions (e.g. proximity ligation assays with RNAPII pS2/PCNA or RNAPII pS2/SAMHD1)?

Minor Points:

1) The yH2AX panels in Fig. 2G seem to be flipped between control and SAMHD1 knockdown cells as based on the images, the shLuc cells would have much higher yH2AX signal.

**Have all data underlying the figures and results presented in the manuscript been provided?**

Reviewer #1: **No: **I have not been provided with an accession code to a genomic site for DRIPseq data presented in the paper.

Reviewer #2: Yes

Reviewer #3: Yes

PLOS authors have the option to publish the peer review history of their article (what does this mean?). If published, this will include your full peer review and any attached files.

Reviewer #1: No

Reviewer #2: No

Reviewer #3: No

---

## [Decision Letter · Decision Letter 1]

24 Feb 2021

Dear Dr Ahn,

Thank you very much for submitting your Research Article entitled 'Aicardi-Goutières syndrome-associated gene SAMHD1 preserves genome integrity by preventing R-loop formation at transcription–replication conflict regions' to PLOS Genetics.

The manuscript was fully evaluated at the editorial level and by independent peer reviewers. The reviewers appreciated the attention to an important topic and the level of revision experiments. However, Reviewer 2 has identified some outstanding concerns that we ask you address in a revised manuscript.

We therefore ask you to modify the manuscript according to the review recommendations. Your revisions should address the specific points made by Reviewer 2.

[LINK]

Yours sincerely,

Natalia Gromak, PhD

Guest Editor

PLOS Genetics

Peter McKinnon

Section Editor: Cancer Genetics

PLOS Genetics

Reviewer's Responses to Questions

**Comments to the Authors:**

Reviewer #2: The authors have made a great effort revising this manuscript, with the addition of numerous new experiments. Overall, these additional data have greatly improved the manuscript, which is now potentially suitable for publication. I still have two concerns and one minor request.

My first concern is that although the authors have provided new evidence that R-loops levels are higher in SAMHD1-depleted cells, it is a bit disappointed that they analyzed only two genes (NEK7 and SEC24D) by DRIP-qPCR.

The second (major) concern relates to the experiment shown in Fig. S4H to support the view that the same replication origins are used in the absence of SAMHD1. The fact that DNA copy number does not change at loci of interest does not mean that the same origins are used. Changes in DNA copy number in a population of cells indicates at best the time of replication (early or late), not origin usage. Moreover, the replication of the reference locus used to normalize DNA copy numbers could also change in SAMHD1-deficient cells, which would affect the interpretation of the data. In short, this experiment is inconclusive and should be removed. Removal of these data would also weaken the argument that origin usage does not change in the absence of SAMHD1 and this limitation needs to be discussed in the manuscript.

Finally, the novel PLA experiments are very nice, but the intensity of PLA foci in Figs 4G and 3A is too low and should be increased.

Reviewer #3: The authors have addressed all my previous comments/concerns and have also added significant new experimental data including DRIP-qPCR validation of the bioinformatic analysis, S9.6 IF and FACS quantifications with appropriate positive and negative controls as well as PLA data between RNAPII-PCNA as requested by the reviewers. Together, the results substantiate the author’s conclusions that R-loops are highly enriched at transcription-replication conflict regions of the genome in fibroblast of patients bearing SAMHD1 mutation. Therefore, I’m positive and support timely publication of the manuscript in Plos Genetics.

**Have all data underlying the figures and results presented in the manuscript been provided?**

Reviewer #2: Yes

Reviewer #3: Yes

PLOS authors have the option to publish the peer review history of their article (what does this mean?). If published, this will include your full peer review and any attached files.

Reviewer #2: No

Reviewer #3: **Yes: **Stephan Hamperl

---

## [Editor Report · Decision Letter 2]

29 Mar 2021

Dear Dr Ahn,

We are pleased to inform you that your manuscript entitled "Aicardi-Goutières syndrome-associated gene SAMHD1 preserves genome integrity by preventing R-loop formation at transcription–replication conflict regions" has been editorially accepted for publication in PLOS Genetics. Congratulations!

Yours sincerely,

Natalia Gromak, PhD

Guest Editor

PLOS Genetics

Peter McKinnon

Section Editor: Cancer Genetics

PLOS Genetics

Comments from the reviewers (if applicable):

**Data Deposition**

http://datadryad.org/submit?journalID=pgenetics&manu=PGENETICS-D-20-01541R2

**Press Queries**

---

## [Editor Report · Acceptance letter]

8 Apr 2021

PGENETICS-D-20-01541R2 

Aicardi-Goutières syndrome-associated gene SAMHD1 preserves genome integrity by preventing R-loop formation at transcription–replication conflict regions 

Dear Dr Ahn, 

We are pleased to inform you that your manuscript entitled "Aicardi-Goutières syndrome-associated gene SAMHD1 preserves genome integrity by preventing R-loop formation at transcription–replication conflict regions" has been formally accepted for publication in PLOS Genetics! Your manuscript is now with our production department and you will be notified of the publication date in due course.

With kind regards,

Katalin Szabo

PLOS Genetics

On behalf of:
